# Validation of video analysis of marker-less barbell auto-tracking in weightlifting

**Hideyuki Nagao** [1]☯*, **Daichi Yamashita** [2]☯

**1** Department of Sports Research, Japan Institute of Sport Sciences, Kita-ku, Tokyo, Japan, **2** Department of Sport Science, Japan Institute of Sport Sciences, Kita-ku, Tokyo, Japan

☯ These authors contributed equally to this work.
* hideyuki.nagao@jpnsport.go.jp

## Abstract

We determined the marker-less barbell auto-tracking accuracy using the Kanade–Lucas–Tomasi (KLT) algorithm in a digital video for two-dimensional analysis (2D-AT). The position coordinates of the barbell's right end during multiple loads (60%–90% of one-repetition maximum) of snatch motion in eight participants were recorded using a three-dimensional motion capture system. Simultaneously, the snatch motion was recorded by a digital camera from the right side. Based on the digital video, the center of the barbell's right end was auto-tracked using the KLT algorithm. Six barbell kinematic variables (forward, backward, and vertical displacement, peak forward, backward, and vertical velocity) were calculated. Intra-class correlation coefficient (ICC) analysis was performed to establish the agreement level between the methods. The Bland–Altman plots and regression were used to examine the agreement between the methods. ICCs of 0.999–0.971 revealed a strong agreement level between the methods. The Bland–Altman plot showed small bias (-0.001–0.001 m, -0.034—0.005 m/s). We considered that 2D-AT could obtain barbell position coordinates with sufficient accuracy to discriminate the difference in the lifter's level and a successful or unsuccessful lift.

## Introduction

The barbell trajectory during lifting motion needs to be analyzed to improve weightlifting performance. Previous studies have reported that the barbell forward displacement in the 2nd pull position to the most forward position of the snatch was 0.037 m, and the peak horizontal speed in the forward direction was 0.25 m/s larger in inferior lifters than that of superior lifters [1]. Furthermore, the barbell backward displacement after the 2nd pull phase was 0.056 m larger in the inferior lifter than that of the superior lifter [2]. An analysis of the barbell trajectory shows the difference between successful and unsuccessful lifts as well as the level of competition in weightlifting. Specifically, the horizontal displacement of the barbell was 0.01 m larger in a successful snatch than that in an unsuccessful snatch [3]. Non-national level lifters reported a 0.01–0.02 m decrease in the amount of the horizontal displacement of the barbell before and after the practice of snatch [4]. A previous study indicated the need for coaches and scientists to monitor barbell kinematic variables because such variables are correlated with weightlifting performance [5].

**Data Availability Statement:** All relevant data are within the manuscript and its Supporting Information files.

**Funding:** H.Nagao number:18K17847 Japan Society for the Promotion of Science KAKENHI https://www.jsps.go.jp/j-grantsinaid/.

A linear position transducer (LPT) is a device used for obtaining data on barbell kinematic and kinetic variables. LPTs use a cable attached to a barbell, which measures variables using a potentiometer or rotary encoder [6]. One of the advantages of using an LPT is that the output of the variables is immediate and can be utilized by one person without taking up space. However, since the LPT is a device that measures the length of the drawn cable, the output is limited to one-dimensional data [7]. In weightlifting, the factors related to the performance and success of the trial include the peak and average vertical velocity, barbell power, and horizontal displacement and velocity of the barbell [1–3]. Hence, the information obtained from the LPT is insufficient. To utilize the scientific knowledge for weightlifting training, it is necessary to obtain data regarding the variables in the horizontal direction.

Three-dimensional (3D) motion-capture (3D-MC) systems are currently the gold standard to obtain an object's position coordinates with high accuracy. However, the 3D-MC system is very expensive, requires a large space, and takes considerable time to be prepared. Actually, the system must be simpler to analyze the barbell trajectory in training and competitions. An analysis method using digital video data taken with one digital video camera has been proposed as a simple method for analyzing the barbell trajectory during the lifting motion [8]. In this system, a light-emitting diode (LED) sensor is attached to the end of the barbell, the lifting motion is recorded with a video camera, and the position coordinates of the LED are automatically tracked based on the digital video data. The LED sensor has a very light weight (37 g) and can be easily attached to a barbell; therefore, it is expected to provide good applicability in weightlifting practice. However, there is an issue concerning durability, as LED sensors are damaged when the barbell falls to the ground. Furthermore, in weightlifting competitions, these sensors may not be attached to the barbell.

The direct linear transformation method has been used in previous studies for calibration in analysis methods using a digital video camera [9,10]. Nonetheless, this method is not suitable for analysis in competitions because it is necessary to enter the movement space. Indeed, the limitations of manual digitization include time and digitization error. Moreover, the calibration method using the known barbell plate diameter (0.45 m) has been used in previous studies [3,11,12], although the accuracy remains uncertain.

Therefore, we propose a marker-less auto-tracking system for barbell trajectory using the Kanade–Lucas–Tomasi (KLT) algorithm [13,14] from digital video for two-dimensional analysis (2D-AT), using the calibration from the barbell plate diameter (0.45 m) as the reference. Compared with previous methods, 2D-AT can contribute to the analysis of weightlifting performance by expanding the range of barbell position coordinates. Weightlifting is an indoor competition, and the lifters do not move too far from the spot. Therefore, since the camera can be fixed and the video can be recorded without significant changes in the light environment, the color and shape of the target to be tracked are considered to be less varied. Therefore, the position of the barbell during weightlifting is considered to be suitable for automatic digitization by the KLT algorithm. From these, we hypothesized that the KLT algorithm could quantify with sufficient accuracy the variables related to the barbell trajectory that determine differences in weightlifting performance. In particular, when the barbell trajectory during lifting can be objectively evaluated during competitions, it will provide important information for athletes and coaches, which could be used in training.

## Materials and methods

### Participants

Eight men participated in this investigation (height: 166.5 ± 4.0 cm, body mass: 74.3 ± 9.7 kg, snatch one-repetition maximum: 105.3 ± 6.9 kg). The participants had at least 3 years of

experience in resistance training and the snatch exercise. They were not active weightlifters; however, they were weightlifting coaches. The study was approved by the Institutional Review Board of the Japan Institute of Sports Sciences (approval number: 2019034). All participants were informed of the benefits and risks of the investigation prior to signing an institutionally approved informed consent document to participate in the study.

## Procedures

A barbell (Eleiko Sport AB, Halmstad, Sweden) with a length of 2.2 m and a weight of 20 kg, approved by the International Weightlifting Federation (IWF), was used for the experiments. The plates attached to the barbell (Uesaka, Tokyo, Japan) were also approved by the IWF. To verify the accuracy of the method, it was necessary to provide some variation in independent variables [15]. Therefore, the load of the snatch was set to multiple conditions of 60%, 70%, 80%, and 90% of one-repetition maximum. A total of 160 snatches were recorded, whereas every subject completed five repetitions for each load condition. Preliminary experiments showed that a load of <50% of one-repetition maximum would result in an unnatural snatch motion; therefore, loads within that range were not included in the present study. In addition, the experiment with the maximum lifting weight was not employed to reduce the risk of injury. In the experiment, the warming-up method and the rest time between lifting were decided by each participant.

The position coordinates of the 14-mm-diameter reflective marker attached to the barbell during the snatch motions were recorded at 600 Hz using an eight-camera 3D-MC system (Vicon MX; Vicon Motion Systems, Oxford, UK). Two reflective markers were placed at each of the barbell's right and left ends, resulting in a total of four markers (Fig 1). In previous studies, the sampling frequency was at 200–250 Hz in the 3D motion analysis of snatch [1,5,16]. However, the body collides with the barbell during the second pull phase of the snatch [17]. Therefore, the barbell position was measured at a higher sampling frequency in the present study (600 Hz) than those in the previous studies (200–250 Hz) to accurately measure the trajectory of the barbell, including the vibration due to the collision.

To confirm the accuracy of the 3D-MC system, the length of the barbell was calculated from the left and right points of the barbell recorded by the 3D-MC system. The distance between the midpoint of the two markers on the right end of the barbell and the midpoint of the two markers on the left end of the barbell was taken as the length of the barbell. The mean difference in the barbell length against 2.2 m was 0.41 ± 0.32 mm (maximum: 1.10 mm, minimum: <0.01 mm).

The lifting motion was simultaneously recorded using a digital camera (ILCE-7M3; Sony, Tokyo, Japan) and telephoto zoom lens (SEL70200G; Sony, Tokyo, Japan) with the 3D-MC system. The accuracy was verified by comparing 2D-AT with the gold standard—that is, 3D-MC. The digital video data were recorded in color at a resolution of 1080 × 1920 (full high definition), approximately 100 Mbps, at a sampling rate of 100 Hz, using a shutter speed of 1/500 s, and using a white balance setting of 5500 K. The digital camera (image sensor) was placed orthogonal to the sagittal plane on the right side of the participant. Following the suggestion of a previous study [10], the digital camera (image sensor) was positioned at 20 m from the barbell's right end and 0.70 m in height. The field of view of the digital video was based on the size of the weightlifting platform (width: 4 m, depth: 4 m) used in competitions organized by the IWF. The focus of the digital camera was set in a manual adjusting mode, and the focus was adjusted at the barbell's right end. All experiments were conducted in the laboratory, not at the competition venue.

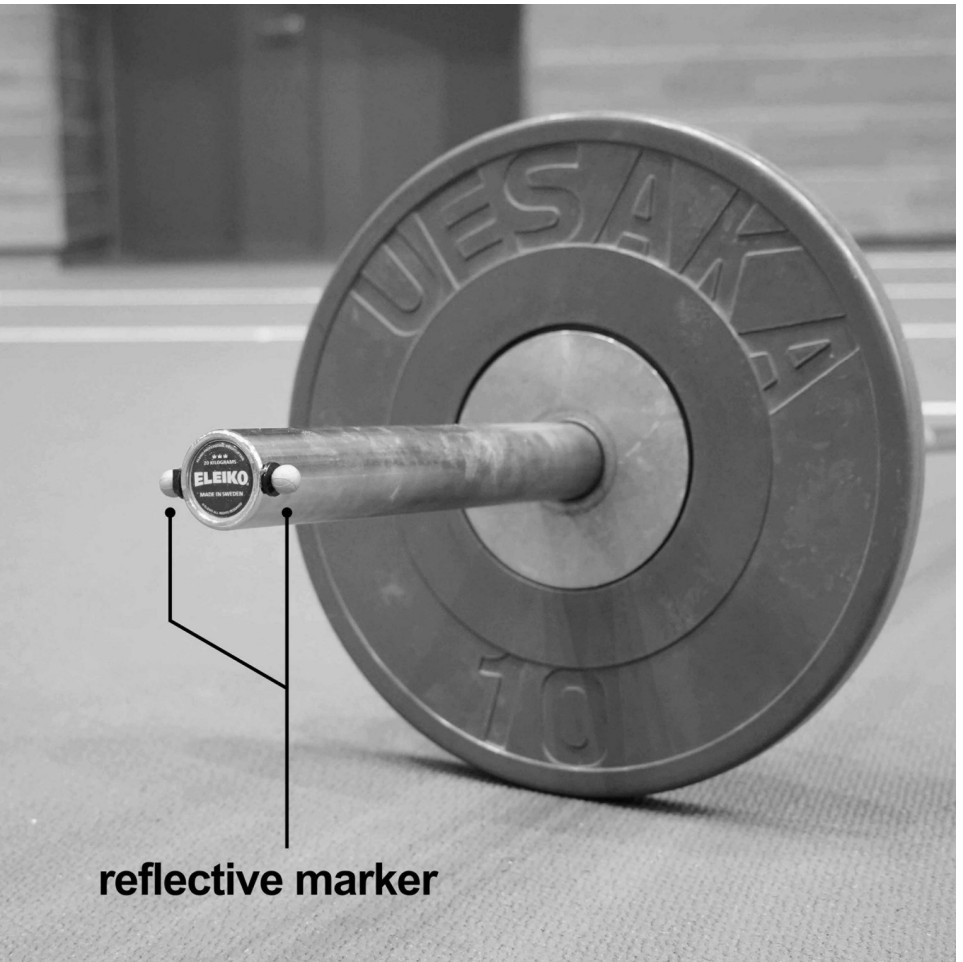

**Fig 1. Placement points of the reflective markers on the barbell's right end.**

A still image of a digital video taken with a digital camera is shown in Fig 2. Based on the digital video, the center of the barbell's right end was auto-tracked using the KLT algorithm [13,14]. There are several methods for computer-based auto-tracking; nevertheless, in this study, we employed the KLT algorithm, which is the basis for these methods [18]. The KLT algorithm tracks where a point in an image has moved in the next frame based on the local color gradient in the image. There is an open-source of the KLT algorithm in numerous computer languages. We developed a custom script written in Mathematica version 11.3 (Wolfram Research Inc., Champaign, IL, USA) using the built-in function "ImageFeatureTrack" [19,20]. The start points of auto-tracking (center of the barbell's right end) were manually selected for the digital video in each attempt. Based on a previous study [3,11,12], the barbell plate diameter (0.45 m) was used as the reference for calibrating the barbell's real-space from the camera-space position coordinates. The diameter of the plate was converted from its width and height (pixel) as 0.45 m by manually obtaining the positional coordinates of the right and left ends of the plate and the top and bottom ends of the plate.

The mid-point between the position coordinates of the two markers on the right end of the bar was obtained with 3D-MC and was used as the barbell position coordinates. The 2D plane in the participant's sagittal plane was analyzed. The $x$-axis indicated the front-rear direction, whereas the $y$-axis indicated the vertical direction. The direction of the lifter's line of sight was

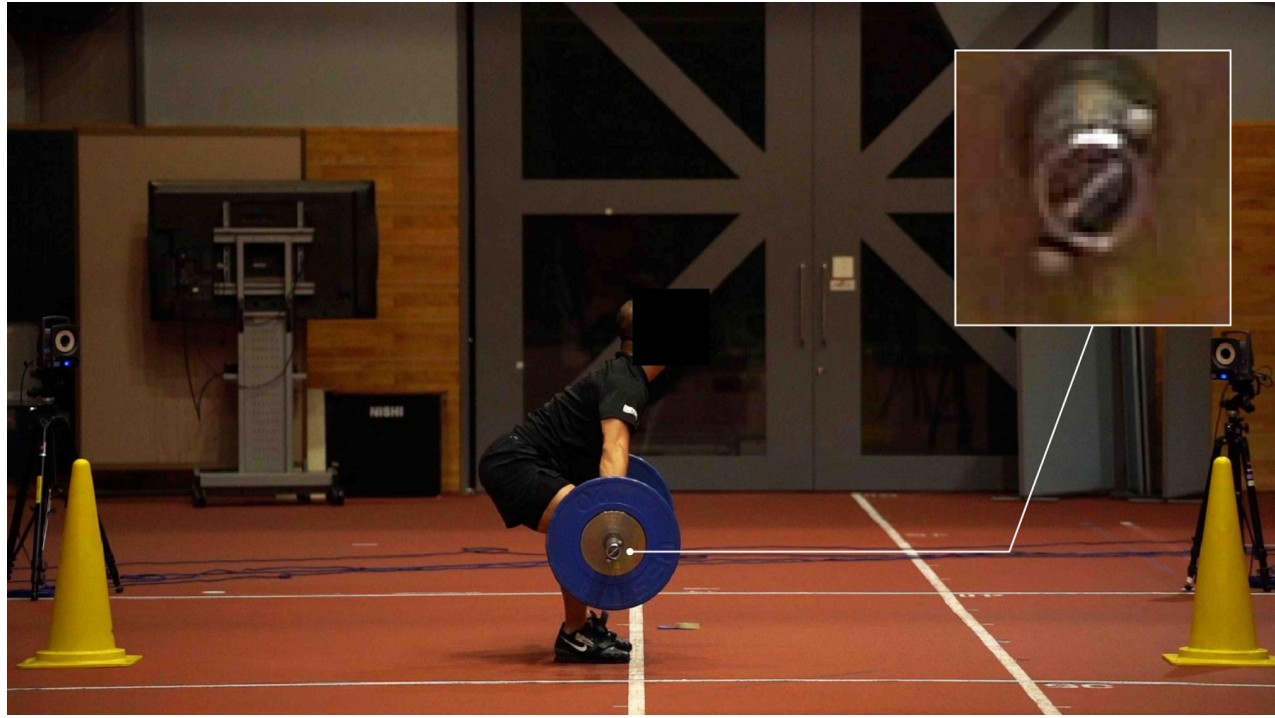

**Fig 2. One of the video images for the marker-less auto-tracking.**

set in a positive direction, and the definition of the coordinate system was the same for the data obtained with 2D-AT. The position coordinates of the barbell's right end obtained with 3D-MC and 2D-AT were smoothed using a fourth-order Butterworth low-pass filter, with the optimum cut-off frequency calculated using the residual analysis method [21]. The cut-off frequency for the $x$-axis and $y$-axis of 3D-MC were 46.0 ± 5.2 Hz and 50.7 ± 5.0 Hz, respectively, whereas the corresponding values of 2D-AT were 13.6 ± 1.0 Hz and 11.6 ± 0.5 Hz, respectively.

Six barbell kinematic variables used in previous studies [1–3] for the analysis of the snatch were calculated from the smoothed 2D position coordinates of the barbell's right end obtained with 3D-MC and 2D-AT. Forward displacement (DxF), backward displacement (DxB), maximum vertical height (DyMH), maximum horizontal linear velocity in the forward direction (pVxF), maximum horizontal linear velocity in the backward direction (pVxB), and maximum vertical linear velocity (pVy) were defined and calculated for each lift. A central difference approximation was used to calculate the velocity. The "start position" and "catch position" (analysis range) were defined based on the methods of a previous study [3]. The displacement and velocity in the horizontal and vertical directions, which are widely used in the biomechanical analysis of snatches, were compared between the methods. All analyses were performed using a personal computer (MacBook Pro with macOS Mojave, running on 2.5 GHz Intel Core i7 and 16 GB 2133 MHz LPDDR3; Apple Inc., Cupertino, CA, USA) and Mathematica version 11.3 (Wolfram Research Inc., Champaign, IL, USA).

## Statistical analyses

Regression analysis with Pearson's product-moment correlation coefficient (r) was performed to investigate the linear relationship between the barbell kinematic variables calculated from

3D-MC and 2D-AT. Intraclass correlation coefficient (ICC, 2.1) analysis was employed to examine the level of agreement between methods. Bland–Altman plots were used to identify potential systematic bias and standard deviations as indicators of precision. Precision is independent of the true value and is a measure of the degree of statistical variance among the measured values. The statistical significance level was set at $p < 0.05$. All statistical analyses were performed using Mathematica version 11.3 (Wolfram Research Inc., Champaign, IL, USA).

## Results

Representative examples of the barbell's 2D trajectory, position coordinates, and velocity obtained with 3D-MC and 2D-AT are presented in Fig 3. The trajectory of the barbell in the snatch usually has an S-shaped pattern [22,23]. The participants in this study also showed an S-shaped pattern. Therefore, it is considered that the participants had the essential skills of the snatch. The time required for 2D-AT per trial was approximately 30 s (0.15 s per frame), and it took 60–90 s per trial to copy the digital video data to a personal computer, auto-track, and calculate each variable.

The results of the 3D-MC and 2D-AT correlation analysis for six variables calculated from 160 trials are presented in Fig 4. If the plotted point is above the identity line, the 2D-AT indicates that the variable has been overestimated. In all variables, a significant positive relationship between the 3D-MC and 2D-AT was observed. Furthermore, there was a very high agreement between the two methods and the variables, as revealed by the ICC (DxF: 0.997, 95% CI 0.994–0.998; DxB: 0.993, 95% CI 0.990–0.995; DyMH: 0.999, 95% CI 0.999–0.999; pVxF: 0.944, 95% CI 0. 894–0. 967; pVxB: 0.995, 95% CI 0.987–0.997; pVy: 0.995, 95% CI 0.993–0.997).

Table 1 presents the systematic errors by the Bland–Altman analysis. The precision was almost the same among all the displacement-related variables (DxF, DxB, and DyMH); however, among the velocity-related variables, pVxF showed a relatively larger value than pVxB and pVy. This indicates that pVxF has a relatively larger statistical variability in the measurement results among the variables of velocity in the 2D-AT method. The line of equality was within the agreement limits (Fig 5). Concerning the limits of agreement (LoA in Fig 5), the method of this study showed better results than those of previous studies that used other methods (smartphone app, LPT) to obtain barbell velocity and displacement [24,25]. The limits of

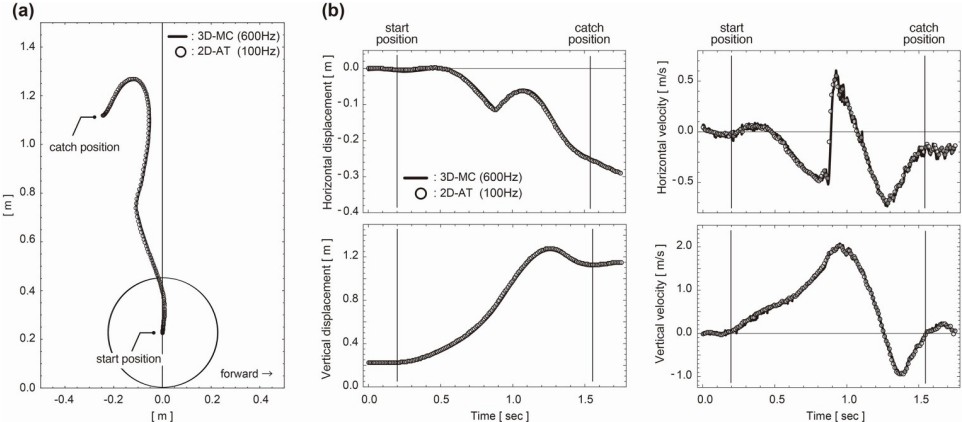

**Fig 3.** A representative bar trajectory of snatch (a) and horizontal and vertical position coordinates and velocity (b) The thick line represents the data from the 3D-MC system, whereas the circle indicates the data from 2D-AT. The direction of the lifter's line of sight was set in the forward direction 3D-MC: Three-dimensional motion capture system, 2D-AT: Two-dimensional analysis.

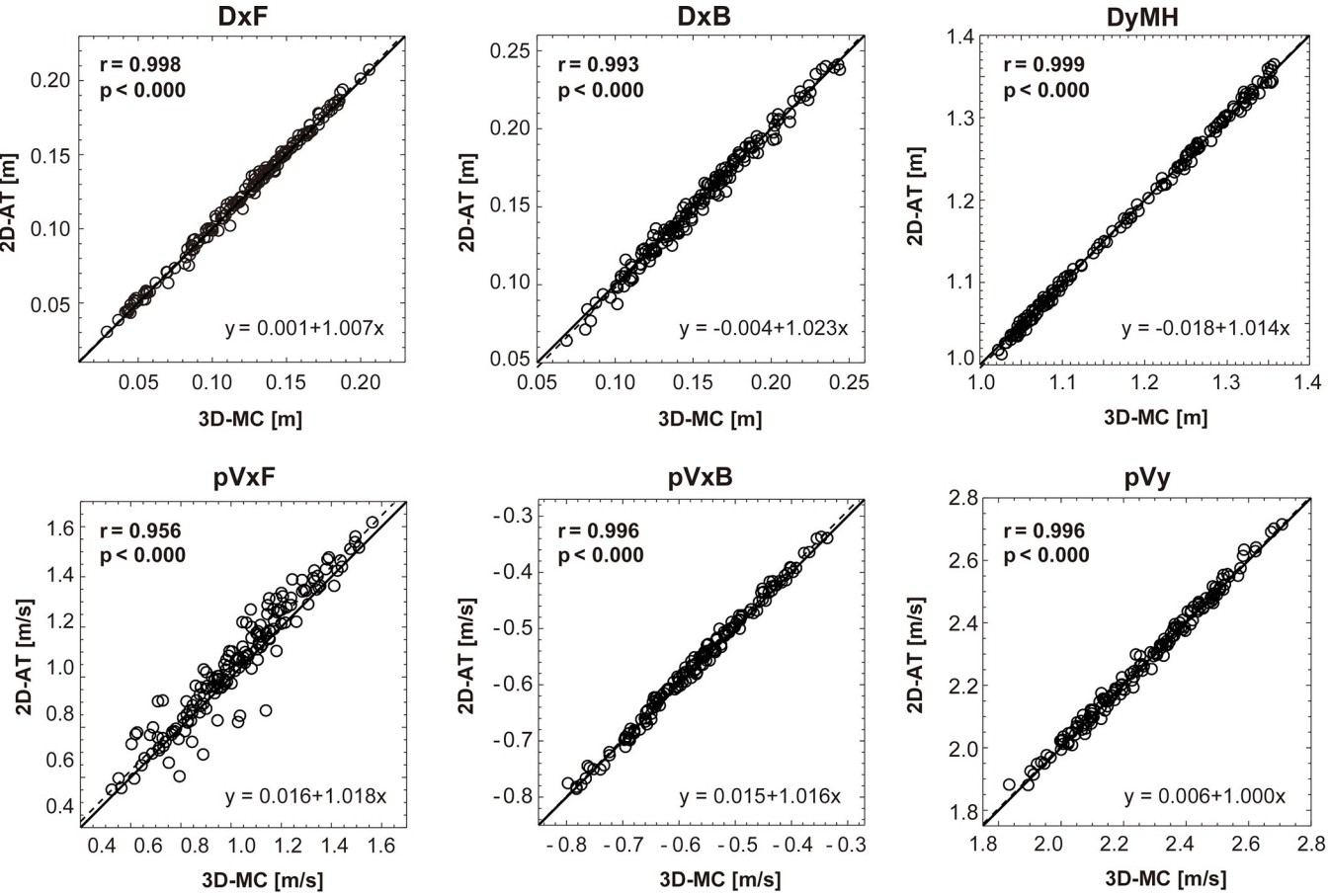

**Fig 4. Linear relationship between the data from the 3D-MC and 2D-AT methods.** The dashed and the thick lines correspond to the linear regression and identity lines, respectively 3D-MC: Three-dimensional motion capture system, 2D-AT: Two-dimensional analysis. DxF: The displacement of forward direction, DxB: The displacement of backward direction, DyMH: The maximum vertical height, pVxF: The maximum horizontal linear velocity in the forward direction, pVxB: The maximum horizontal linear velocity in the backward direction, pVy: Maximum vertical linear velocity.

agreement are for visual judgment of how well two methods of measurement agree. The smaller the range between these two limits, the better the agreement is.

## Discussion

This study was designed to validate kinematic data derived from 2D-AT in comparison with 3D-MC. The primary finding was that the analysis derived from the KLT algorithm showed

**Table 1. Mean and standard deviation (std) values of variables, bias, and precision between the two methods in the Bland–Altman analysis.**

| variable | unit | mean ± std 3D-MC | mean ± std 2D-AT | bias (3D-MC–2D-AT) | precision |
|---|---|---|---|---|---|
| DxF | [m] | 0.121 ± 0.041 | 0.122 ± 0.041 | -0.001 | 0.003 |
| DxB | [m] | 0.155 ± 0.037 | 0.155 ± 0.038 | 0.001 | 0.004 |
| DyMH | [m] | 1.180 ± 0.110 | 1.178 ± 0.112 | 0.001 | 0.005 |
| pVxF | [m/s] | 1.023 ± 0.224 | 1.057 ± 0.238 | -0.034 | 0.071 |
| pVxB | [m/s] | -0.568 ± 0.101 | -0.562 ± 0.103 | -0.005 | 0.009 |
| pVy | [m/s] | 2.266 ± 0.190 | 2.272 ± 0.191 | -0.006 | 0.018 |

3D-MC: Data from 3D motion capture system, 2D-AT: Data from out-tracing with the video movie.

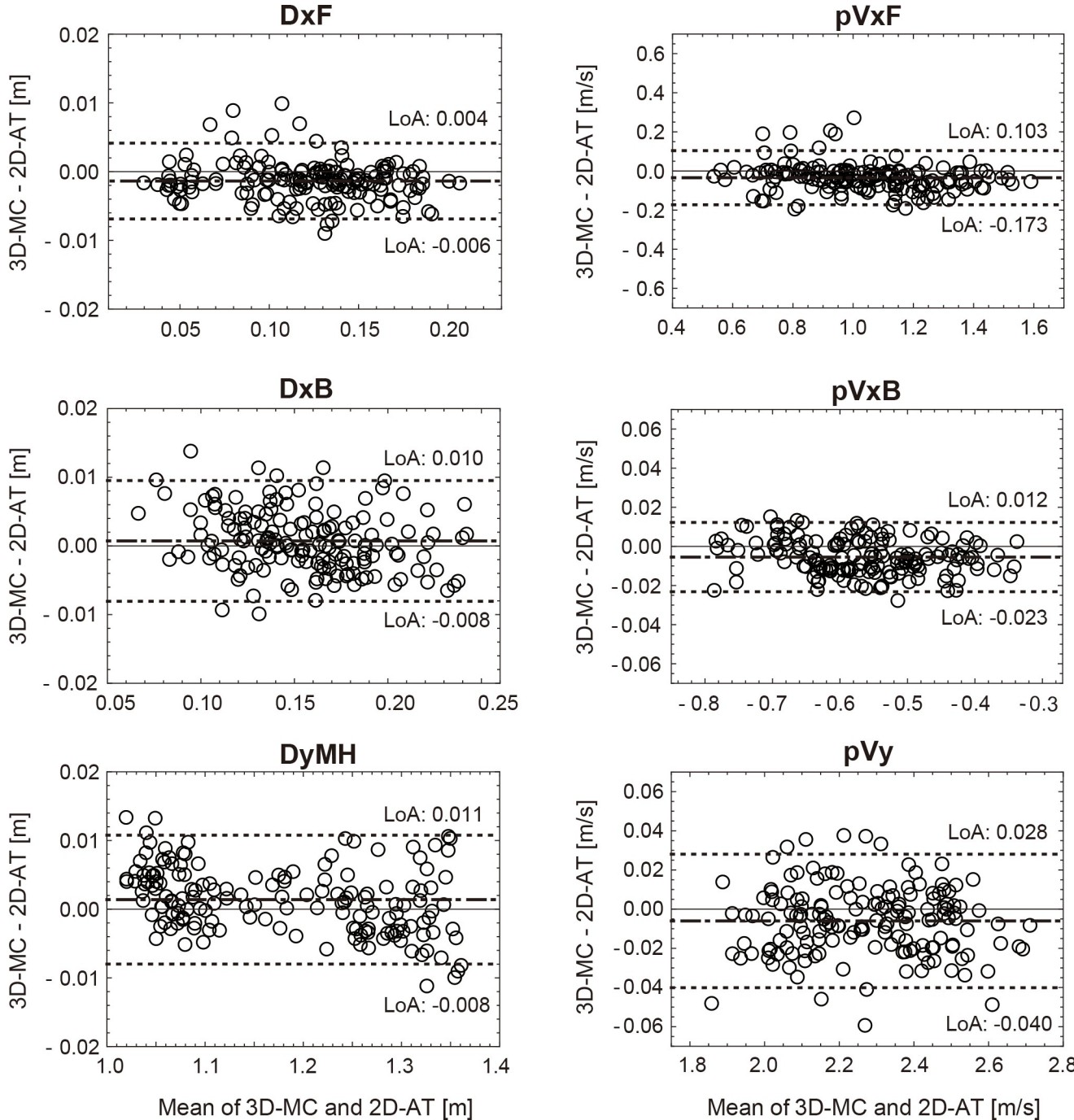

**Fig 5. The Bland–Altman plot of the mean differences across the 3D-MC and 2D-AT methods.** The data are plotted against the mean value for both methods (dot-dashed line), with the upper and lower 95% LoA shown as dotted lines 3D-MC: Three-dimensional motion capture system, 2D-AT: Two-dimensional analysis, LoA: Limits of agreement. DxF: The displacement of forward direction, DxB: The displacement of backward direction, DyMH: The maximum vertical height, pVxF: The maximum horizontal linear velocity in the forward direction, pVxB: The maximum horizontal linear velocity in the backward direction, pVy: Maximum vertical linear velocity.

high concurrent validity when compared with laboratory-based instrumentation. The results of the correlation analysis for the 3D-MC and 2D-AT showed significant correlations for all variables (r > 0.90). Furthermore, the line of equality was within the agreement limits in the

Bland–Altman plot. For weightlifting analysis, the 3D-MC system is costly and requires a large space to place several cameras. It was considered that by employing the 2D-AT system that needs only one digital camera, it was possible to obtain the barbell trajectory at a lower cost than the 3D-MC system, and without the limitations of location, while maintaining the accuracy of the measurement.

A previous study reported a significant difference of 0.056 m for the barbell backward displacement (DxB in this report) and of approximately 0.1 m (0.086–0.109 m) for the maximum height (DyMH) in the snatch between the superior and inferior lifters [2]. Furthermore, DxB is significantly greater by 0.01 m for successful attempts than that for failed attempts at snatch [3]. Using the auto-tracking by the KLT algorithm shown in this study, the bias of the barbell displacement variable was lower than 0.01 m for DxF, DxB, and DyMH (Table 1). The bias of the variables for barbell displacement presented in this study was smaller than the differences in barbell variables between athletes with different lifting performances and between successful and unsuccessful snatches presented in previous studies. Therefore, it is considered that 2D-AT can obtain the barbell position coordinates with sufficient accuracy to quantify differences in the barbell displacement between successful and unsuccessful lifts as well as between different levels of lifters.

With respect to the variables related to barbell velocity, it was shown that the value of pVxF (barbell maximum forward velocity) was significantly lower by 0.25 m/s in the lifter who lifted a higher weight in the competition than that of the lifter who lifted a lower weight [1]. A previous study pointed out that the allowable error for barbell velocity in weightlifting analysis is 0.03 m/s [26]. In this study, the bias of pVxF and pVxB was -0.034 m/s and 0.005 m/s, respectively (Table 1), which was sufficiently smaller than the magnitude of the velocity difference attributable to the competition level. However, the absolute value of the bias of pVxF was >0.03 m/s. Furthermore, the range of LoA of pVxF in the Brand-Altman plot was ± 0.138 m/s (-0.173 to 0.103 m/s), which is approximately 10 times larger than that of the other variables. From these results, it is considered that 2D-AT can obtain the pVxB with sufficient accuracy for weightlifting analysis. However, barbell forward velocity may not provide data with sufficient accuracy for the analysis of weightlifting. With respect to pVy, no significant difference between the performance level of lifting and success or failure was noted [1–3]; however, similar to the other variables, the magnitude of bias was <1% of the mean value of the variables (-0.006 m/s).

The coefficient of determination of pVxF was relatively small compared with those of other variables (Fig 4). The precision in the Bland–Altman analysis also indicated a relatively large value (Table 1). In snatch, the bar comes into full contact with the body at the crease of the hips at approximately the time when the trunk reaches at a vertical orientation. When the barbell contacts the body significantly earlier, it will gain excessive horizontal force [17]. As pVxF appears immediately after the barbell contacts the body, the magnitude of pVxF is considered to include the low-frequency component of vibration in the *x*-axis component of the barbell position coordinates. Therefore, the difference in sampling frequency may have affected the magnitude of the first-order differentiated value.

A previous study [8] proposed an auto-tracking method by attaching a LED to the end of the barbell and increasing the contrast with the background of the barbell end on the image. However, with the 2D-AT method shown in this study, it is not necessary to attach a sensor or device to the barbell, and the lifter's motion can be analyzed in an environment where nothing changes from the usual. However, our study had some limitations. Specifically, we used a barbell that had a dark blue end (Fig 1). Interestingly, the barbell designs and colors vary according to the manufacturer. The KLT algorithm tracks a target point based on the local color gradient in the image. Therefore, the accuracy of tracking could have been slightly different in

the case of different color barbells. In particular, barbells used during women's competitions almost always have a yellow end. Those barbells have lower contrast than the barbell used during men's competitions that was employed in this study, which has a dark blue end. The effect of the barbell's condition on the auto-tracking with the KLT algorithm, including color differences and the presence or absence of reflective markers, needs to be investigated in the future. In this study, the camera was placed orthogonal to the sagittal plane of the lifter and 20 m from the end of the barbell. However, this condition may not be replicated depending on the environment of the competition venue or training space. The effect of the distance and angle of the camera placement from the object being measured on the accuracy of the auto-tracking with the KLT algorithm of the barbell is a topic for future research. In addition, if it becomes clear how the resolution and sampling frequency of the video data affect the accuracy of the auto-tracking with the KLT algorithm of the barbell, it is expected that the application range of this method will be further expanded.

## Conclusions

When the position coordinates of the barbell trajectory can be obtained, the lifting techniques can be objectively evaluated by the kinematic variables during weightlifting practice or competition. By employing the marker-less auto-tracking shown in this study, it was possible to analyze the barbell trajectory with high accuracy. However, there are some limitations in the accuracy of the barbell forward velocity. This method is low in cost and has few restrictions on tools and places; hence, it can be used to analyze weightlifting in various places, such as competition venues and practice areas. The method proposed in this study has the potential to be applied in sagittal plane exercises using barbells, such as squats.

## Supporting information

**S1 Table. Barbell kinematics data.** This is the file containing the data used for the statistical analyses in this paper.
(CSV)

## Author Contributions

**Conceptualization:** Hideyuki Nagao, Daichi Yamashita.

**Investigation:** Hideyuki Nagao, Daichi Yamashita.

**Methodology:** Hideyuki Nagao.

**Project administration:** Hideyuki Nagao, Daichi Yamashita.

**Supervision:** Hideyuki Nagao.

**Validation:** Hideyuki Nagao.

**Writing – original draft:** Hideyuki Nagao.

**Writing – review & editing:** Hideyuki Nagao, Daichi Yamashita.

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
