## [Decision Letter · Decision Letter 0]

5 Mar 2021

PONE-D-20-38952

Validation of Video Analysis of Marker-Less Barbell Auto-Tracking in Weightlifting

PLOS ONE

Dear Dr. Nagao,

Thank you for submitting your manuscript to PLOS ONE. After careful consideration, we feel that it has merit but does not fully meet PLOS ONE’s publication criteria as it currently stands. Therefore, we invite you to submit a revised version of the manuscript that addresses the points raised during the review process.

We look forward to receiving your revised manuscript.

Kind regards,

Yumeng Li

Academic Editor

PLOS ONE

Additional Editor Comments:

Some major revisions are needed based on reviewers' comments

Journal Requirements:

2. Please provide further details on sample size and power calculations.

3. In statistical methods, please refer to any post-hoc corrections to correct for multiple comparisons during your statistical analyses. If these were not performed please justify the reasons. Please refer to our statistical reporting guidelines for assistance (https://journals.plos.org/plosone/s/submission-guidelines.#loc-statistical-reporting).

4. In your statistical analyses, please state whether you accounted for repeated measurements per participant.

"The authors wish to thank the International Weightlifting Federation and the Japan Weightlifting

 Association for their support of this study."

"H.Nagao

number:18K17847

Japan Society for the Promotion of Science KAKENHI

" ext-link-type="uri" xlink:type="simple">https://www.jsps.go.jp/j-grantsinaid/"

Reviewers' comments:

Reviewer's Responses to Questions

**Comments to the Author**

1. Is the manuscript technically sound, and do the data support the conclusions?

Reviewer #1: Partly

Reviewer #2: Yes

2. Has the statistical analysis been performed appropriately and rigorously? 

Reviewer #1: N/A

Reviewer #2: Yes

3. Have the authors made all data underlying the findings in their manuscript fully available?

Reviewer #1: Yes

Reviewer #2: Yes

4. Is the manuscript presented in an intelligible fashion and written in standard English?

Reviewer #1: Yes

Reviewer #2: No

5. Review Comments to the Author

Reviewer #1: I would like to compliment the authors on their research paper. In my opinion, the methodological approach is described transparently, designed thoroughly and it arises logically from the discussed rationale of simplifying video analysis in weightlifting. The analytical approach is similar to recently published validation studies on technology for kinematic assessment. However, a few points need to be considered to ensure a high academic quality of the paper:

- Line 8: Please reconsider the spelling of the word “peek”. Do you mean “peak” (maximum)?

- Line 24: For clarity, I suggest specifying whether the reported change was an increase or decrease in horizontal displacement.

- 32-33: Currently, there is a number of LPT systems available that simultaneously assesses the angle of the drawn cable, allowing to calculate the tether position either in 2 dimensions (e.g. GymAware, Kinetic Performance Technology) or 3 dimensions (e.g. RepOne, Red Matter Labs, Inc). Hence, I suggest removing or reformulating this sentence.

- Line 77-78: I suggest reformulating the sentence: “[…] 3 years of experience in resistance training and the snatch exercise.”

- Line 155: Apparently, statistical analysis was performed on a pooled data set (i.e. all 160 repetitions were included at once). As stated by Orange et al. (2019) this approach may be common in validation studies, yet it violates the assumption of independence of data. This should particularly be considered since the present study only features eight subjects. To assure that the nested (pooled) structure of data does not influence the results, I suggest recalculating the main statistics using a multilevel approach treating subjects as s random variable. If this multilevel approach does not substantially change the statistics, the presentation of your current results (using the pooled data) would be inherently justified as an equivalently valuable, but simpler model.

- Line 161: Although it is widely applied in validation research, I recommend against interpreting the proportionality effect in Bland-Altman analyses. It has been suggested that Bland-Altman analysis succumbs artifactual bias, in that it shows systematic proportional bias in controlled simulations, where no systematic bias is prevalent (Hopkins, 2004). A better solution would be to interpret the slope coefficient of your regression analysis as a measure of proportional error. If it is significantly different from 1 (not from zero!) this could be interpreted as a systematic proportional error. However, this issue may not be too problematic in the present investigation, as there seems to be only little random error. Hence, the artifactual bias from Bland-Altman analysis can probably be neglected.

- Line 205 (Table 1): Please specify for the reader what the statistic “precision” means. Also, I wonder if you applied any form of statistical correction to your calculation of bias precision? If I recalculate them as the simple mean of differences and the standard deviation of differences between 3D-MC and 2D-AT using your provided data file, my result for bias in pVxF (-0.034 m/s) and my result for precision in pVxB (0.009 m/s) differ slightly from your reported values. Interestingly, only those two variables show noticeable differences from your values in my calculations. I may be wrong, but please check the calculation on those statistics, as any error might influence your interpretation.

- Line 241-245: You interpret your result according to a sample mean bias. For pVxF, the magnitude is indeed below 0.03 m/s, but your LOAs range from -0.173 to 0.103 m/s. This random variability in accuracy should not be neglected in your interpretation! Here is why: According to a simple descriptive calculation I did on you provided data file, only about 40% of your sample’s pVxF values actually achieve an accuracy that falls within your proposed range of +/- 0.03 m/s. If you want to include magnitude thresholds like 0.03 m/s in your paper, this should be included in your statistical analysis, e.g. by applying equivalence testing (Lakens, 2017). However, there are various ways to analytically include uncertainty. At the very least, you should consider confidence intervals in the interpretation.

- Line 254: Please maintain a consistent style of referencing according to the journal’s guidelines.

Hopkins, W.G. (2004) Bias in Bland-Altman but not Regression Validity Analyses. Sportscience 8, 42-46. (sportsci.org/jour/04/wghbias.htm)

Lakens D. Equivalence Tests: A Practical Primer for t Tests, Correlations, and Meta-Analyses. Soc Psychol Personal Sci. 2017 May;8(4):355-362. doi: 10.1177/1948550617697177

Orange ST, Metcalfe JW, Liefeith A, Marshall P, Madden LA, Fewster CR, Vince RV. Validity and Reliability of a Wearable Inertial Sensor to Measure Velocity and Power in the Back Squat and Bench Press. J Strength Cond Res. 2019 Sep;33(9):2398-2408. doi: 10.1519/JSC.0000000000002574

Reviewer #2: Ref #: PONE-D-20-38952.

TITLE: VALIDATION OF VIDEO ANALYSIS OF MARKER-LESS BARBELL AUTO-TRACKING IN WEIGHTLIFTING.

SUMMARY: This is a topical study that is of interest to Plos One readership however the paper lacks clarity in key areas. The practical relevance of this work is not clear. The authors are urged to rewrite the Introduction to provide greater clarity as to the practical relevance of their work. The authors should discuss technical determinants of successful performance in weightlifting. For instance, why is it important to track barbell trajectory in training / competition? What are the major issues with current methods? These points should be clarified to the reader. The Introduction contains detail that is relevant to the Methods section. This should be addressed to aid interpretation. Research Methods and statistical tests used are appropriate to answer the research question proposed. Greater clarity is required in parts to facilitate reproducibility of the study.

ABSTRACT: Page 2, Line 8; Replace period with comma after displacement.

Page 2, Line 13; The authors should clarify better the practical relevance of their findings. The authors state that 2D-AT could obtain barbell position coordinates with sufficient accuracy to discriminate the difference in barbell velocity due to the lifting performance difference. The meaning of this point is unclear. Are they referring to differences between analysis methods, between exercises or between levels of lifters?

INTRODUCTION: Page 3, Line 19; The authors provide data illustrating differences between inferior and superior lifters. The authors should be clear throughout when making such comparisons. Did superior or inferior lifters exhibit more or less horizontal displacement? This section would benefit from a sentence that clarifies technical determinants of successful performance in weightlifting.

Page 3, Line 21; The authors state that there was a 0.25 m/s difference in bar linear velocity. Between which groups? Which population exhibited greater velocities?

Page 3, Line 23; Correct 0.0244 to two decimal places.

Page 3, Line 25; Why should analysis of barbell trajectory during lifting enhance performance? What is the basis for this?

Page 3, Line 29; Check wording here. Consider rewording to something like ‘LPTs use a cable attached to a barbell which measures variables using a potentiometer or rotary encoder.’

Page 3, Line 34; The authors state determinants of success in weightlifting performance. This should be stated earlier in the introduction to give context to discussions on the importance of measuring bar trajectory.

Page 3, Line 39; Change ‘system’ to ‘systems’ and replace ‘is’ with ‘are’.

Page 4 Line 49; Replace ‘by’ with ‘when’.

Page 4, Line 59; The authors state that “The accuracy was verified by comparing 2D-AT with the gold standard—that is, 3D-MC”. Are they referring to the current study or to previous studies? If referring to previous studies, please include relevant citations. If referring to the current study, please save this information for the Methods section.

Page 4, Line 61 – 68; This detail should be removed from the Introduction and placed in the Methods section.

METHODS: Page 5, Line 77; replace ‘year’ with ‘years’. Since the snatch is a technical lift, the level of participants’ proficiency with this exercise should be clarified for the reader. Did participants have three years of experience in the performance of the snatch? Were participants competitive lifters or did they perform the snatch in training on occasion?

Page 5, Line 79; Please clarify if all participants were informed of the benefits of the study as opposed to just a single participant. Reword this sentence to clarify.

Page 5, Line 84; The authors state the barbell used was dedicated for men. Please classify the bar based on its weight and dimensions rather than gender.

Page 5, Line 88; Greater clarity required here. How many repetitions did participants perform at each load? How much recovery did participants receive between repetitions within and between loads? Why did the authors choose the loads presented? Considering the intended application of the current data to competition and the use of maximum loads during competition, is the data valid under one-repetition maximum conditions? Please clarify.

Page 5, Line 89; What is meant by the “The warming-up method and the rest time between lifting were free”? Did participants complete their own desired warm-up routine? Please reword.

Page 6, Line 102; The authors state that two reflective markers were placed at the end of the barbell. This information is also stated in line 93. Please clarify how many markers were attached the barbell and digitized? This should be stated once only.

Page 6, Line 103; It is stated that barbell length was regulated for competition. Please clarify the location of data collection. Specifically, were data collected in a lab or at a competition venue?

Page 6, Line 104; Change 2,200 mm to 2.2-m.

Page 6, Line 113; Insert ‘the’ before ‘weightlifting platform’.

Page 7, Line 135; Insert ‘was’ after ‘bar’.

Page 7, Line 144; Insert ‘the’ after ‘of.

RESULTS: Page 8, Line 167; Insert ‘the’ before ‘snatch’.

Page 8, Line 168; Replace ‘participant’ with ‘participants’.

Page 8, Line 169; Replace ‘participant’ with ‘participants’.

Page 8, Line 183; The authors present ICC data describing the level of agreement between the two methods. Please include also the 95% confidence interval of the ICC to identify the upper and lower bound intervals that describe the ICC.

Page 9, Line 196; The authors state that “pVxF showed a relatively larger value than pVxB and pVy”. Please simplify this statement to enhance clarity for the reader. Is it meant that the 2D-AT method exhibited greater error for velocity measures relative to the 3D-MC method?

Page 9, Line 199; Please insert a sentence that clarifies the specific nature of the limit-of-agreement to suggest that data showed good agreement between the 2D-AT and 3D-MC methods for kinematic measures.

DISCUSSION: Page 10, Line 218; Delete “measures obtained by”.

Page 10, Line 225; What is meant by having a few restrictions on tools and places? Is it meant that the 2D-AT method can be used with limited equipment in a variety of environmental settings?

Page 10, Line 233; The meaning of this sentence is unclear. Consider rewording.

Page 10, Line 236; Check wording here. Are the authors referring to differences between successful and unsuccessful lifts?

Page 11, Line 239; The authors are urged to be specific throughout to aid clarity. What is meant by high and low competition levels?

Page 11, Line 248; The meaning of this sentence is unclear. Consider rewording.

Page 11, Line 255; Insert ‘magnitude of’ before pVxF and delete ‘value magnitude’.

Page 12, Line 267; Please change the reference to ‘women’s barbells’ to ‘barbells used during women’s competitions’. Please also refer to ‘barbell’s used during men’s competitions’.

6. PLOS authors have the option to publish the peer review history of their article (what does this mean?). If published, this will include your full peer review and any attached files.

Reviewer #1: **Yes: **Benedikt Mitter

Reviewer #2: No

---

## [Author Response · Author response to Decision Letter 0]

19 May 2021

Response to Reviewer #1

Comment 1:

Line 8: Please reconsider the spelling of the word “peek”. Do you mean “peak” (maximum)?

Response:

We apologize for the spelling error. The original manuscript has been revised as per the reviewers’ comment.

Abstract, Page 2, Line 8

peek → peak 

Comment 2:

Line 24: For clarity, I suggest specifying whether the reported change was an increase or decrease in horizontal displacement.

Response:

Thank you for your helpful comments. Based on the reference, the manuscript has been revised as follows:

Introduction, Page 3, Line 24

Added: Non-national level lifters reported a 0.01-0.02 m decrease ...

Comment 3:

32-33: Currently, there is a number of LPT systems available that simultaneously assesses the angle of the drawn cable, allowing to calculate the tether position either in 2 dimensions (e.g. GymAware, Kinetic Performance Technology) or 3 dimensions (e.g. RepOne, Red Matter Labs, Inc). Hence, I suggest removing or reformulating this sentence.

Response:

Based on the latest information about the available LPT systems and the reviewers' suggestions, some parts of the manuscript have been deleted.

Introduction, Page 3

Deleted: Therefore, the output is limited to the peak and average velocity and the barbell vertical component power.

Comment 4:

Line 77-78: I suggest reformulating the sentence: “[…] 3 years of experience in resistance training and the snatch exercise.”

Response:

Thank you for the suggestion. The manuscript has been revised as per the reviewer’s suggestion.

Materials and Methods, Page 5, Line 74

Before: The participants had at least 3 year of resistance training and snatch.

After: The participants had at least 3 years of experience in resistance training and the snatch exercise.

Comment 5:

Line 155: Apparently, statistical analysis was performed on a pooled data set (i.e. all 160 repetitions were included at once). As stated by Orange et al. (2019) this approach may be common in validation studies, yet it violates the assumption of independence of data. This should particularly be considered since the present study only features eight subjects. To assure that the nested (pooled) structure of data does not influence the results, I suggest recalculating the main statistics using a multilevel approach treating subjects as s random variable. If this multilevel approach does not substantially change the statistics, the presentation of your current results (using the pooled data) would be inherently justified as an equivalently valuable, but simpler model.

Response:

In this study, the eight subjects are not independent variables. Studies examining the accuracy of several training-related measures have also shown correlation and Bland-Altman analyses based on multiple data from a single subject [A–D]. Therefore, the issue of sample independence, which is a concern of the reviewers, does not seem to arise.

A) Balsalobre-Fernández, C., Marchante, D., Muñoz-López, M., and Jiménez, S. L. Validity and reliability of a novel iPhone app for the measurement of barbell velocity and 1RM on the bench-press exercise. J. Sports Sci, 2017, 18, 1–7. 

B) Balsalobre-Fernández C, Geiser G, Krzykowski J, Kipp K. Validity and reliability of a computer-vision-based smartphone app for measuring barbell trajectory during the snatch. J Sports Sci. 2020, 38, 710–6.

C) Pérez-Castilla A, Piepoli A, Delgado-García G, Garrido-Blanca G, García-Ramos A. Reliability and concurrent validity of seven commercially available devices for the assessment of movement velocity at different intensities during the bench press. J Strength Cond Res. 2019, 33: 1258–1265.

D) Rider BC, Conger SA, Ditzenberger GL, Besteman SS, Bouret CM, Coughlin AM. Examining the accuracy of the Polar A360 monitor. J Strength Cond Res. 2020. doi: 10.1519/JSC.0000000000003136.

Comment 6:

Line 161: Although it is widely applied in validation research, I recommend against interpreting the proportionality effect in Bland-Altman analyses. It has been suggested that Bland-Altman analysis succumbs artifactual bias, in that it shows systematic proportional bias in controlled simulations, where no systematic bias is prevalent (Hopkins, 2004). A better solution would be to interpret the slope coefficient of your regression analysis as a measure of proportional error. If it is significantly different from 1 (not from zero!) this could be interpreted as a systematic proportional error. However, this issue may not be too problematic in the present investigation, as there seems to be only little random error. Hence, the artifactual bias from Bland-Altman analysis can probably be neglected.

Response:

Thank you for your constructive comment. In line with the reviewer’s comments, the description of the proportional error of the Bland–Altman analysis has been deleted from the entire manuscript. In addition, a regression equation has been added to Figure 1.

Figure 1

Materials and Methods, Page 8 Line 166

Deleted: … and the coefficient of determination (R2) of the regression line on the Bland–Altman plots was calculated to identify the proportionality effect. 

Results, Page 9, Table 1

Deleted: “R2 (p-value)” in Table 1 and “coefficient of determination (R2) with p-values” in description of Table 1.

Results, Page 9, legend of Table 1

Deleted: and proportional

Results, Page 9 Line 203

Deleted: …coefficient of determination of the regression line on the Bland–Altman plots for all variables was not significant (Table 1), and the…

Results, Page 11 Line 240

Deleted: Additionally, no significant correlation (a proportional error) between the variable size and the difference between the methods was found.

Comment 7: 

Line 205 (Table 1): Please specify for the reader what the statistic “precision” means. Also, I wonder if you applied any form of statistical correction to your calculation of bias precision? If I recalculate them as the simple mean of differences and the standard deviation of differences between 3D-MC and 2D-AT using your provided data file, my result for bias in pVxF (-0.034 m/s) and my result for precision in pVxB (0.009 m/s) differ slightly from your reported values. Interestingly, only those two variables show noticeable differences from your values in my calculations. I may be wrong, but please check the calculation on those statistics, as any error might influence your interpretation.

Response:

The meaning of "precision" has been added in the Materials and Methods section as follows:

Materials and Methods, Page 8, Lines 166-167

Added: ... as precision. Precision is independent of the true value and is a measure of the degree of statistical variance of among the measured values.

As the reviewer has pointed out, there were some errors in the values shown in the original manuscript. However, there are no errors in the raw data spreadsheet submitted with the manuscript. Therefore, the values in Table 1 and related parts of the manuscript have been revised. The recalculation results regarding the bias of pVxF was -0.034 m/s. This value exceeds the error in velocity allowed in the analysis of barbells in weightlifting (0.03 m/s), as pointed out in a previous study [26]. In this relation, some parts of the manuscript have been revised and some additions were made.

Results, Page 9 Table 1

Discussion, Page 11, Lines 253-255

Added: However, the absolute value of the bias of pVxF was more than 0.03 m/s. Furthermore, the range of LoA of pVxF in the Brand-Altman plot was ±0.138 m/s (-0.173 to 0.103), which is about 10 times larger than the other variables.

Discussion, Page 11, Lines 255-267

Before: From these results, it is considered that 2D-AT can obtain the barbell horizontal velocity with sufficient accuracy to discriminate the difference in lifting performance.

After: From these results, it is considered that 2D-AT can obtain the pVxB with sufficient accuracy for weightlifting analysis. However, barbell forward velocity may not provide data with sufficient accuracy for the analysis of weightlifting.

Conclusions, Page 12, Lines 294-295

Added: However, there are some limitations in the accuracy of the barbell forward velocity.

Comment 8: 

Line 241-245: You interpret your result according to a sample mean bias. For pVxF, the magnitude is indeed below 0.03 m/s, but your LOAs range from -0.173 to 0.103 m/s. This random variability in accuracy should not be neglected in your interpretation! Here is why: According to a simple descriptive calculation I did on you provided data file, only about 40% of your sample’s pVxF values actually achieve an accuracy that falls within your proposed range of +/- 0.03 m/s. If you want to include magnitude thresholds like 0.03 m/s in your paper, this should be included in your statistical analysis, e.g. by applying equivalence testing (Lakens, 2017). However, there are various ways to analytically include uncertainty. At the very least, you should consider confidence intervals in the interpretation.

Response:

Thank you for your constructive comment. According to the reviewer’s comments, we have added a consideration of LoA with respect to the bias of pVxF. This revision is also related to Comment 7.

Discussion, Page 11, Lines 255-257

Added: However, the absolute value of the bias of pVxF was more than 0.03 m/s. Furthermore, the range of LoA of pVxF in the Brand-Altman plot was ±0.138 m/s (-0.173 to 0.103 m/s), which is about 10 times larger than the other variables.

Comment 9: 

Line 254: Please maintain a consistent style of referencing according to the journal’s guidelines.

Response:

The reference citation has now been revised according to the journal's guidelines.

Discussion, Page 11, Line 265

Before: (Everett et al., 2008)

After: [17]

Response to Reviewer #2

Comment 1:

Page 2, Line 8; Replace period with comma after displacement.

Response:

Thank you for pointing this out. We have revised this as suggested. 

Comment 2:

Page 2, Line 13; The authors should clarify better the practical relevance of their findings. The authors state that 2D-AT could obtain barbell position coordinates with sufficient accuracy to discriminate the difference in barbell velocity due to the lifting performance difference. The meaning of this point is unclear. Are they referring to differences between analysis methods, between exercises or between levels of lifters?

Response:

We have revised the sentences which the reviewer pointed out as unclear as follows:

Abstract, Page 2, Lines 13-14

Before: We considered that 2D-AT could obtain barbell position coordinates with sufficient accuracy to discriminate the difference in the barbell velocity due to the lifting performance difference.

After: We considered that 2D-AT could obtain barbell position coordinates with sufficient accuracy to discriminate the difference in the lifter's level and a successful or unsuccessful lift.

Comment 3:

Page 3, Line 19; The authors provide data illustrating differences between inferior and superior lifters. The authors should be clear throughout when making such comparisons. Did superior or inferior lifters exhibit more or less horizontal displacement? This section would benefit from a sentence that clarifies technical determinants of successful performance in weightlifting.

Response: 

Thank you for your comment. We have added some information about inferior and superior lifters and successful performance in weightlifting.

Introduction, Page 3, Lines 18-23:

Added: Previous studies have reported that the barbell backward displacement in the first pull phase was 0.037 m and the peak horizontal velocity in the forward direction was 0.35 m/s larger in superior lifters compared to inferior lifters [1]. Furthermore, the barbell backward displacement after the second pull phase was 0.056 m larger in the inferior lifter than the superior lifter [2]. An analysis of the barbell trajectory shows the difference between successful and unsuccessful lifts, as well as the level of competition in weightlifting.

Introduction, Page 3, Lines 24-26:

Added: Non-national level lifters reported a 0.01-0.02 m decrease in the amount of the horizontal displacement of the barbell before and after the practice of snatch [4]

Comment 4:

Page 3, Line 21; The authors state that there was a 0.25 m/s difference in bar linear velocity. Between which groups? Which population exhibited greater velocities?

Response:

Based on the reviewer’s comments, we have added details on what variables differed between the groups and how. The order of the references has also been changed.

Introduction, Page 3, Lines 18-21

Before: Previous studies reported a 0.023-0.056 m difference in the amount of horizontal displacement of the barbell during the snatch between an inferior and a superior lifter in national level athletes [1, 2].

After: Previous studies have reported that the barbell backward displacement in 1st pull phase was 0.037 m and the peak horizontal velocity in the forward direction was 0.35 m/s larger in superior lifters compared to inferior lifters [1]. Furthermore, the barbell backward displacement after the seconnd pull phase was 0.056 m larger in the inferior lifter than the superior lifter [2].

Reference, Page 15. and all relevant parts of the manuscript.

[1] → [2], [2] → [1]

Comment 5:

Page 3, Line 23; Correct 0.0244 to two decimal places.

Response:

The manuscript was revised according to the reviewer’s suggestion.

Introduction, Page 3, Line 24

0.0244 → 0.02 

Comment 6:

Page 3, Line 25; Why should analysis of barbell trajectory during lifting enhance performance? What is the basis for this?

Response:

Since the reference did not clearly include evidence as to why the analysis of barbell trajectory during lifting enhances performance, the text was revised as follows:

Introduction, Page 3, Lines 26-27

Before: Interestingly, the analysis of the kinematic and kinetic variables of the barbell trajectory during the lifting motion in competitions and training, and their application in coaching can aid in improving the weightlifting performance [5].

After: A previous study indicated the need for coaches and scientists to monitor barbell kinematic variables because such variables are correlated with weightlifting performance [5].

Comment 7:

Page 3, Line 29; Check wording here. Consider rewording to something like ‘LPTs use a cable attached to a barbell which measures variables using a potentiometer or rotary encoder.’

Response:

The manuscript was revised according to the reviewer’s suggestion.

Introduction, Page 3, Lines 29-30

Before: In LPT, a cable is attached to a barbell, and various variables are calculated from a potentiometer or rotary encoder [6].

After: LPTs use a cable attached to a barbell which measures variables using a potentiometer or rotary encoder [6].

Comment 8:

Page 3, Line 34; The authors state determinants of success in weightlifting performance. This should be stated earlier in the introduction to give context to discussions on the importance of measuring bar trajectory.

Response:

Based on the reviewer's suggestion, a statement about the difference between successful and unsuccessful lifts has been added to the first paragraph of the Introduction. The parts of Line 34 in the original manuscript have not been changed in order to maintain the context of the sentence. 

Introduction, Page 3, Lines 21-22

Added: An analysis of the barbell trajectory shows the difference between successful and unsuccessful lifts, as well as the level of competition in weightlifting. Specifically, …

Revised: Furthermore → Specifically

Comment 9:

Page 3, Line 39; Change ‘system’ to ‘systems’ and replace ‘is’ with ‘are’.

Response:

The manuscript was revised according to the reviewer’s suggestion.

Comment 10:

Page 4 Line 49; Replace ‘by’ with ‘when’.

Response:

The manuscript was revised according to the reviewer’s suggestion.

Comment 11:

Page 4, Line 59; The authors state that “The accuracy was verified by comparing 2D-AT with the gold standard—that is, 3D-MC”. Are they referring to the current study or to previous studies? If referring to previous studies, please include relevant citations. If referring to the current study, please save this information for the Methods section.

Response:

The sentences that were pointed out by the reviewer have been moved to the Materials and Methods section. Also, some sentences were deleted to keep the context after the changes.

Introduction, Page 3 → Materials and Methods, Page 6, Line 111

Moved: The accuracy was verified by comparing 2D-AT with the gold standard—that is, 3D-MC.

Deleted: This study aimed to clarify the accuracy of 2D-AT.

Comment 12:

Page 4, Line 61 – 68; This detail should be removed from the Introduction and placed in the Methods section.

Response:

The sentences that were pointed out by the reviewer have been moved to the Materials and Methods section.

Introduction, Page 3 → Materials and Methods, Page 7, Lines 155-157

Moved: The displacement and velocity in the horizontal and vertical directions, widely used in the biomechanical analysis of snatches, were compared between the methods.

Comment 13:

Page 5, Line 77; replace ‘year’ with ‘years’. Since the snatch is a technical lift, the level of participants’ proficiency with this exercise should be clarified for the reader. Did participants have three years of experience in the performance of the snatch? Were participants competitive lifters or did they perform the snatch in training on occasion?

Response:

Thank you for your helpful comments. Based on the comments of several reviewers, the sentence has been revised as follows:

Materials and Methods, Page 5, Lines 73-75

Before: The participants had at least 3 year of resistance training and snatch.

After: The participants had at least 3 years of experience in resistance training and the snatch exercise. They were not active weightlifters, but they were weightlifting coaches.

Comment 14:

Page 5, Line 79; Please clarify if all participants were informed of the benefits of the study as opposed to just a single participant. Reword this sentence to clarify.

Response:

Based on the comments of several reviewers, the sentence has been revised as follows:

Materials and Methods, Page 5, Line 76

Before: The participant was informed…

After: All participants were informed…

Comment 15:

Page 5, Line 84; The authors state the barbell used was dedicated for men. Please classify the bar based on its weight and dimensions rather than gender.

Response:

Thank you for your helpful comments. Based on the reviewer’s comments, the manuscript has been revised as follows:

Materials and Methods, Page 5, Lines 81-82

Before: A barbell (Eleiko Sport AB, Halmstad, Sweden) dedicated for men and approved by the International Weightlifting Federation (IWF) was used for the experiments.

After: A barbell (Eleiko Sport AB, Halmstad, Sweden) with a length of 2.2 m and a weight of 20 kg, approved by the International Weightlifting Federation (IWF), was used for the experiments.

Comment 16 and 17:

Page 5, Line 88; Greater clarity required here. How many repetitions did participants perform at each load? How much recovery did participants receive between repetitions within and between loads? Why did the authors choose the loads presented? Considering the intended application of the current data to competition and the use of maximum loads during competition, is the data valid under one-repetition maximum conditions? Please clarify.

Page 5, Line 89; What is meant by the “The warming-up method and the rest time between lifting were free”? Did participants complete their own desired warm-up routine? Please reword.

Response:

We apologize for the lack of clarity in the sentence of the manuscript. Criteria for selecting loads for experiments and rest methods have been added and revised to the manuscript as follows:

Materials and Methods, Page 5, Lines 87-90

Added: Preliminary experiments showed that a load of less than 50% of one-repetition maximum would result in an unnatural snatch motion due to low-load, therefore it was excluded from the measurement conditions. In addition, the experiment with the maximum lifting weight was not employed to avoid risks. In the experiment, a maximum of five consecutive snatches was allowed.

Materials and Methods, Page 5, Lines 90-91

Before: The warming-up method and the rest time between lifting were free.

After: The warming-up method and the rest time between lifting were decided by each participant.

Comment 18:

Page 6, Line 102; The authors state that two reflective markers were placed at the end of the barbell. This information is also stated in line 93. Please clarify how many markers were attached the barbell and digitized? This should be stated once only.

Response:

Thank you for your helpful comments. Based on the reviewer’s comments, the manuscript has been revised as follows:

Materials and Methods, Page 5, Lines 94-95

Before: Two reflective markers were placed at the barbell’s right end (Fig 1).

After: Two reflective markers were placed at each of the barbell’s right and left ends, resulting in a total of four markers (Fig 1).

Materials and Methods, Page 6, Lines 104-107

Deleted: … two reflective markers recorded at 600 Hz were also attached to the barbell’s left end.

Added: …the length of the barbell was calculated from the left and right points of the barbell recorded by the 3D-MC system. The distance between the midpoint of the two markers on the right end of the barbell and the midpoint of the two markers on the left end of the barbell was taken as the length of the barbell.

Materials and Methods, Page 6, Lines 107-108

Before: The mean difference in the distance between the left and right markers (barbell length) against 2,200 mm…

After: The mean difference in the barbell length against 2.2 m…

Comment 19:

Page 6, Line 103; It is stated that barbell length was regulated for competition. Please clarify the location of data collection. Specifically, were data collected in a lab or at a competition venue?

Response:

In accordance with the revision of Comment 15, the part pointed out by the reviewer above has been deleted from the manuscript because its contents were overlapping. In addition, a sentence has been added about the place (laboratory) where the measurement was performed.

Materials and Methods, Page 6, Lines 107

Deleted: The length of the barbell for the competition was regulated at 2,200 mm.

Materials and Methods, Page 6, Lines 119-120

Added: All experiments were conducted in the laboratory, not at the competition venue.

Comment 20:

Page 6, Line 104; Change 2,200 mm to 2.2-m.

Comment 21:

Page 6, Line 113; Insert ‘the’ before ‘weightlifting platform’

Comment 22:

Page 7, Line 135; Insert ‘was’ after ‘bar’.

Comment 23:

Page 7, Line 144; Insert ‘the’ after ‘of.

Comment 24:

Page 8, Line 167; Insert ‘the’ before ‘snatch’.

Comment 25 and 26:

Page 8, Line 168 and 169; Replace ‘participant’ with ‘participants’.

Response:

The manuscript was revised according to the reviewer’s suggestions in Comments 20-26.

Comment 27:

Page 8, Line 183; The authors present ICC data describing the level of agreement between the two methods. Please include also the 95% confidence interval of the ICC to identify the upper and lower bound intervals that describe the ICC.

Response:

Thank you for your helpful comments. We have added the 95% confidence interval of the ICCs. 

Results, Page 9, Lines 189-191

Added: ICC (DxF: 0.997, 95%CI 0.994–0.998, DxB: 0.993, 95%CI 0.990–0.995, DyMH: 0.999, 95%CI 0.999–0.999, pVxF: 0.944, 95%CI 0. 894–0. 967, pVxB: 0.995, 95%CI 0.987–0.997, pVy: 0.995, 95%CI 0.993–0.997).

Comment 28:

Page 9, Line 196; The authors state that “pVxF showed a relatively larger value than pVxB and pVy”. Please simplify this statement to enhance clarity for the reader. Is it meant that the 2D-AT method exhibited greater error for velocity measures relative to the 3D-MC method?

Response:

Thank you for your constructive comment. According to reviewer’s comments, a description of the precision of pVxF has been added to the manuscript.

Results, Page 9, Lines 203-204

Added: This indicates that the precision of pVxF is lower than that of pVxB and pVy within the 2D-AT method.

Comment 29:

Page 9, Line 199; Please insert a sentence that clarifies the specific nature of the limit-of-agreement to suggest that data showed good agreement between the 2D-AT and 3D-MC methods for kinematic measures.

Response:

Thank you for your constructive comment. According to the reviewer’s comments, a description of the LoA has been added to the manuscript.

Results, Page 9, Lines 207-209

Added: The limits of agreement are for visual judgement of how well two methods of measurement agree. The smaller the range between these two limits the better the agreement is.

Comment 30:

Page 10, Line 218; Delete “measures obtained by”.

Response:

The manuscript was revised according to the reviewer’s suggestion.

Comment 31:

Page 10, Line 225; What is meant by having a few restrictions on tools and places? Is it meant that the 2D-AT method can be used with limited equipment in a variety of environmental settings?

Response:

Thank you for your constructive comment. In order to avoid ambiguity in the sentences, the manuscript was revised as follows:

Discussion, Pages 10-11, Lines 230-234

Before: It was considered that by employing the 2D-AT system, it was possible to analyze the barbell trajectory with high accuracy, low cost, and with only a few restrictions on tools and places.

After: For weightlifting analysis, the 3D-MC system is costly and requires a large space to place many cameras. It was considered that by employing the 2D-AT system that needs only one digital camera, it was possible to obtain the barbell trajectory at a lower cost than the 3D-MC system, and without the limitations of location, while maintaining the accuracy of the measurement.

Comment 32:

Page 10, Line 233; The meaning of this sentence is unclear. Consider rewording.

Response:

The sentences have been revised for clarity as follows:

Discussion, Page 11, Lines 240-243

Before: The magnitude of this bias is considered to be sufficiently smaller than the corresponding of the variable difference indicating the snatch technique, which has been clarified in the previous study.

After: The bias of the variables for barbell displacement presented in this study was smaller than the differences in barbell variables between athletes with different lifting performances and between successful and unsuccessful snatches presented in previous studies.

Comment 33:

Page 10, Line 236; Check wording here. Are the authors referring to differences between successful and unsuccessful lifts?

Response:

In order to avoid ambiguity in the sentences, the manuscript was revised as follows:

Discussion, Page 11, Lines 245-246

Before: …with different performances.

After: …with differences between the performance level of lifting and success or failure.

Comment 34:

Page 11, Line 239; The authors are urged to be specific throughout to aid clarity. What is meant by high and low competition levels?

Response:

For the reader who is not so knowledgeable about weightlifting, the manuscript has been revised as follows:

Discussion, Page 11, Lines 247-249

Before: …previous studies reported that the pVxF (barbell maximum forward velocity) was significantly lower by 0.25 m/s in athletes with high than in those with low competition levels [2].

After: … it was shown that the value of pVxF (barbell maximum forward velocity) was significantly lower by 0.35 m/s in the lifter who lifted a higher weight in the competition than the lifter who lifted a lower weight [1].

Comment 35:

Page 11, Line 248; The meaning of this sentence is unclear. Consider rewording.

Response:

Due to other revisions that increased the word count of the manuscript, unclear sentences that had been pointed out as low priority were deleted from the manuscript.

Discussion, Page 11, Line 260

Deleted: As multiple loads of snatches have been analyzed and samples of variables were widely obtained, the results of this study could be applied even if a lifting technique were to be developed in the future.

Comment 36:

Page 11, Line 255; Insert ‘magnitude of’ before pVxF and delete ‘value magnitude’.

Response:

The manuscript was revised according to the reviewer’s suggestion.

Comment 37:

Page 12, Line 267; Please change the reference to ‘women’s barbells’ to ‘barbells used during women’s competitions’. Please also refer to ‘barbell’s used during men’s competitions’.

Response:

The manuscript was revised according to the reviewer’s suggestion as follows:

Discussion, Page 12, Lines 277-279

Before: In particular, women's barbells for competition almost always have a yellow end. This is a lower contrast than the dark blue (men's barbells for competition).

After: In particular, barbells used during women’s competitions almost always have a yellow end. This is a lower contrast than the barbell used during men’s competitions that was employed in this study, which has a dark blue end.

---

## [Decision Letter · Decision Letter 1]

24 Aug 2021

PONE-D-20-38952R1

Validation of Video Analysis of Marker-Less Barbell Auto-Tracking in Weightlifting

PLOS ONE

Dear Dr. Nagao,

Thank you for submitting your manuscript to PLOS ONE. After careful consideration, we feel that it has merit but does not fully meet PLOS ONE’s publication criteria as it currently stands. Therefore, we invite you to submit a revised version of the manuscript that addresses the points raised during the review process.

If applicable, we recommend that you deposit your laboratory protocols in protocols.io to enhance the reproducibility of your results. Protocols.io assigns your protocol its own identifier (DOI) so that it can be cited independently in the future. For instructions see: http://journals.plos.org/plosone/s/submission-guidelines#loc-laboratory-protocols. Additionally, PLOS ONE offers an option for publishing peer-reviewed Lab Protocol articles, which describe protocols hosted on protocols.io. Read more information on sharing protocols at https://plos.org/protocols?utm_medium=editorial-emailutm_source=authorlettersutm_campaign=protocols.

We look forward to receiving your revised manuscript.

Kind regards,

Yumeng Li

Academic Editor

PLOS ONE

Journal Requirements:

Additional Editor Comments (if provided):

Some minor revisions are still needed before acceptance.

Reviewers' comments:

Reviewer's Responses to Questions

**Comments to the Author**

1. If the authors have adequately addressed your comments raised in a previous round of review and you feel that this manuscript is now acceptable for publication, you may indicate that here to bypass the “Comments to the Author” section, enter your conflict of interest statement in the “Confidential to Editor” section, and submit your "Accept" recommendation.

Reviewer #1: All comments have been addressed

Reviewer #2: (No Response)

2. Is the manuscript technically sound, and do the data support the conclusions?

Reviewer #1: Yes

Reviewer #2: Yes

3. Has the statistical analysis been performed appropriately and rigorously? 

Reviewer #1: Yes

Reviewer #2: Yes

4. Have the authors made all data underlying the findings in their manuscript fully available?

Reviewer #1: Yes

Reviewer #2: Yes

5. Is the manuscript presented in an intelligible fashion and written in standard English?

Reviewer #1: Yes

Reviewer #2: Yes

6. Review Comments to the Author

Reviewer #1: I believe that by considering the reviewers’ suggestions, the authors managed to eliminate or explain unclear statements and therefore substantially improved the comprehensibility of their article for future readers. That said, I still noticed a few points that I would like to address in order to have the article fulfill a high scientific standard.

Line 18-20: I’d like to encourage the authors to change the wording and ensure that the numbers from the addressed study (Ikeda et al., 2012 [1]) are reported correctly. First, I suggest providing information on what exercise you are addressing (the snatch).

Second, in terms of rewording, it is not perfectly clear if the reported 0.037 m of barbell backward displacement refers to superior lifters, inferior lifters, the difference between superior and inferior lifters or lifters in general. Also, the statement “[…] peak horizontal velocity in the forward direction was 0.35 m/s larger in superior lifters compared to inferior lifters” could be misleading to some readers, since it may suggest higher horizontal movement speed in superior lifters. That, however, would not be true since the mean values reported in the referenced paper of Ikeda et al. (2012) were -0.38 m/s for “Best Lifter” and -0.63 m/s for “Japanese Lifter” (p. 1287, Table 4), so the “Best Lifter” group showed a lower magnitude of speed. Knowing that the paper expresses velocity in forward direction in negative values is therefore crucial to interpret your statement correctly. I suggest using a much simpler approach and directly address “speed” (a scalar, hence omitting information on direction) rather than “velocity” (a vector) in your statement. Consider using something like “[…] the peak horizontal speed in the forward direction was … m/s higher in inferior lifters compared to superior lifters”.

Third, in terms of revising the numbers of your statement: I could not find the reported values of 0.037 m and 0.35 m/s in the paper of Ikeda et al. (2012) for the associated variables in your manuscript. A value of 0.037 m would fit the difference of group means for the variable Dx3 (0.040 and 0.077), but according to the paper, this variable represents the horizontal displacement in the interval “Second pull position to the most forward position”, not the first pull phase. Wouldn’t the first pull phase be equivalent to the variable Dx1 from the referenced paper? Concerning peak horizontal velocity in forward direction, the difference in group means (I already stated the numbers reported in the paper above) does not yield 0.35, but 0.25 m/s. Please ensure that the numbers you are referencing (also from other papers) are adopted correctly, especially when you include them in the interpretation of your results (e.g. in the discussion, Line 246-248).

Line 47: Consider rewording: “[…] it is expected to provide good applicability in weightlifting practice” rather than “[…] it is expected to be used in weightlifting practice situations”

Line 82-83: Consider rewording: “The plates attached […] were also approved […]” rather than “The plate attached […] was also approved […]”

Line 86: The sentence “A total of 160 snatches were recorded 5 times for each load” is a bit irritating. Consider rewording to something like “A total of 160 snatches were recorded, whereas every subject completed 5 repetitions for each load condition.”

Line 86-88: Consider deleting “due to low load” and rewording to “[…], therefore loads within that range were not included in the present study”

Line 89: Consider rewording: “[…] to reduce the risk of injury” rather than “[…] to avoid risks”

Line 89-90: The expression “[…], a maximum of five consecutive snatches was allowed” is not clear to me. Did participants have the option to perform less than 5 snatches per load? If are you addressing the time in between lifts at the same load, I’m not sure if this information is necessary, given that you state in the following sentence that participants were free to choose their rest time.

Line 211-213 (Table 1): Thank you for correcting the values on the Bland-Altman analysis! The values for bias now match my calculations. However, considering that you interpret “precision” as the standard deviation of differences between the two methods, the precision value for pVxB is still different in my calculations. My result would yield a precision for pVxB of 0.009 (in comparison you report a precision of 0.001). Again, I may be wrong, but given that our calculations match for all the other statistics, I suggest you should revise your calculation in this specific case.

Line 269: Please change “[…] attaching an LED […]” to “[…] attaching a LED […]”

Reviewer #2: SUMMARY: The authors have addressed the majority of my comments. Some minor issues remain with wording in parts. Please see specific comments below.

METHODS:

Page 5, Line 86; check wording of this sentence. Is it true that participants performed 160 snatches at each load? The authors should reword to clarify the specific number of repetitions completed by each participant at each load.

Page 5, Line 93; insert ‘system’ after 3D-MC.

Page 5, Line 97; the authors state that barbell position was measured at “a higher sampling frequency”. Please clarify the exact sampling frequency to aid repeatability of the study.

Page 5, Line 113; insert ‘at a’ before ‘sampling’. Insert ‘using a’ before ‘shutter’. Reword ‘and white balance’ with ‘using a white balance setting’.

RESULTS

Page 9, Line 203; The authors state that “pVxF showed a relatively larger value than pVxB and pVy”. Please simplify this statement to enhance clarity for the reader. Is it meant that the 2D-AT method exhibited greater error for velocity measures relative to the 3D-MC method? This has not been addressed since the original review.

DISCUSSION: Page 11, Line 242; the authors should be more concise. Consider rewording to something like ‘Therefore, it is considered that 2D-AT can obtain the barbell position coordinates with sufficient accuracy to quantify differences in the barbell displacement between successful and unsuccessful lifts as well as between different levels of athlete’.

7. PLOS authors have the option to publish the peer review history of their article (what does this mean?). If published, this will include your full peer review and any attached files.

Reviewer #1: **Yes: **Benedikt Mitter

Reviewer #2: No

---

## [Author Response · Author response to Decision Letter 1]

2 Sep 2021

Response to Reviewer #1

Comment 1:

Line 18-20: I’d like to encourage the authors to change the wording and ensure that the numbers from the addressed study (Ikeda et al., 2012 [1]) are reported correctly. First, I suggest providing information on what exercise you are addressing (the snatch).

Second, in terms of rewording, it is not perfectly clear if the reported 0.037 m of barbell backward displacement refers to superior lifters, inferior lifters, the difference between superior and inferior lifters or lifters in general. Also, the statement “[…] peak horizontal velocity in the forward direction was 0.35 m/s larger in superior lifters compared to inferior lifters” could be misleading to some readers, since it may suggest higher horizontal movement speed in superior lifters. That, however, would not be true since the mean values reported in the referenced paper of Ikeda et al. (2012) were -0.38 m/s for “Best Lifter” and -0.63 m/s for “Japanese Lifter” (p. 1287, Table 4), so the “Best Lifter” group showed a lower magnitude of speed. Knowing that the paper expresses velocity in forward direction in negative values is therefore crucial to interpret your statement correctly. I suggest using a much simpler approach and directly address “speed” (a scalar, hence omitting information on direction) rather than “velocity” (a vector) in your statement. Consider using something like “[…] the peak horizontal speed in the forward direction was … m/s higher in inferior lifters compared to superior lifters”.

Third, in terms of revising the numbers of your statement: I could not find the reported values of 0.037 m and 0.35 m/s in the paper of Ikeda et al. (2012) for the associated variables in your manuscript. A value of 0.037 m would fit the difference of group means for the variable Dx3 (0.040 and 0.077), but according to the paper, this variable represents the horizontal displacement in the interval “Second pull position to the most forward position”, not the first pull phase. Wouldn’t the first pull phase be equivalent to the variable Dx1 from the referenced paper? Concerning peak horizontal velocity in forward direction, the difference in group means (I already stated the numbers reported in the paper above) does not yield 0.35, but 0.25 m/s. Please ensure that the numbers you are referencing (also from other papers) are adopted correctly, especially when you include them in the interpretation of your results (e.g. in the discussion, Line 246-248).

Response:

We thank the reviewer for these very insightful comments. We apologize for the incorrect interpretation of the variable Dx3 (the forward barbell displacement in the 2nd second pull position to the most forward position, in Ikeda 2012). This has been revised following the reference. The difference between the Japanese lifter (JL, inferior lifter) and the best lifter (BL, superior lifter) in the peak horizontal barbell velocity in the forward direction has been revised to the correct value (BL: -0.38 vs. JL: -0.63, Absolute value [speed] of JL is greater [0.25] than that of BL). In addition, we have added the information that these variables are related to the snatch.

Introduction, Page 3, Lines 19–20

Before:

Previous studies have reported that the barbell backward displacement in the 1st pull phase was 0.037 m and the peak horizontal velocity in the forward direction was 0.35 m/s larger in superior lifters compared to inferior lifters [1].

After:

Previous studies have reported that the barbell forward displacement in the 2nd pull position to the most forward position of the snatch was 0.037 m, and the peak horizontal speed in the forward direction was 0.25 m/s larger in inferior lifters than that of superior lifters [1].

Discussion, Page 11, Line 249

Before: … pVxF (barbell maximum forward velocity) was significantly lower by 0.35 m/s in the …

After: … pVxF (barbell maximum forward velocity) was significantly lower by 0.25 m/s in the …

Comment 2:

Line 47: Consider rewording: “[…] it is expected to provide good applicability in weightlifting practice” rather than “[…] it is expected to be used in weightlifting practice situations”

Response:

We thank the reviewer for this constructive comment. We have revised the manuscript following your comment.

Introduction Page 4, Line 48

Before: it is expected to be used in weightlifting practice situations.

After: it is expected to provide good applicability in weightlifting practice.

Comment 3:

Line 82-83: Consider rewording: 

“The plates attached […] were also approved […]” rather than “The plate attached […] was also approved […]”

Response:

We thank the reviewer for this important suggestion. We have revised the manuscript following your suggestion.

Materials and Methods Page5, Line 84

Before: The plate attached […] was also approved […]

After: The plates attached […] were also approved […]

Comment 4:

Line 86: The sentence “A total of 160 snatches were recorded 5 times for each load” is a bit irritating. Consider rewording to something like “A total of 160 snatches were recorded, whereas every subject completed 5 repetitions for each load condition.”

Response:

We thank the reviewer for this valuable suggestion. We have revised the manuscript according to your suggestion.

Materials and Methods Page 5, Lines 87–88

Before: A total of 160 snatches were recorded 5 times for each load.

After: A total of 160 snatches were recorded, whereas every subject completed five repetitions for each load condition.

Comment 5:

Line 86-88: Consider deleting “due to low load” and rewording to “[…], therefore loads within that range were not included in the present study”

Response:

We thank the reviewer for this pertinent suggestion. We have revised the manuscript according to your suggestion.

Materials and Methods Page 5, Lines 89–90

Before: … snatch motion due to low-load, therefore it was excluded from the measurement conditions.

After: … snatch motion; therefore, loads within that range were not included in the present study.

Comment 6:

Line 89: Consider rewording: “[…] to reduce the risk of injury” rather than “[…] to avoid risks”

Response:

We thank the reviewer for pointing this out. We have revised the manuscript according to your comment.

Materials and Methods Page 5, Line 91

Before: … not employed to avoid risks.

After: … not employed to reduce the risk of injury.

Comment 7:

Line 89-90: The expression “[…], a maximum of five consecutive snatches was allowed” is not clear to me. Did participants have the option to perform less than 5 snatches per load? If are you addressing the time in between lifts at the same load, I’m not sure if this information is necessary, given that you state in the following sentence that participants were free to choose their rest time.

Response: 

We thank the reviewer for this very valuable comment. Following your comment, we have revised the manuscript to delete unnecessary sentences. 

Materials and Methods Page 5, Lines 91–92

Before: In the experiment, a maximum of five consecutive snatches was allowed. The warming-up method and the rest time between lifting were decided by each participant.

After: In the experiment, the warming-up method and the rest time between lifting were decided by each participant.

Comment 8:

Line 211-213 (Table 1): Thank you for correcting the values on the Bland-Altman analysis! The values for bias now match my calculations. However, considering that you interpret “precision” as the standard deviation of differences between the two methods, the precision value for pVxB is still different in my calculations. My result would yield a precision for pVxB of 0.009 (in comparison you report a precision of 0.001). Again, I may be wrong, but given that our calculations match for all the other statistics, I suggest you should revise your calculation in this specific case.

Response:

We thank the reviewer for this very pertinent suggestion. We had misjudged the calculation result by one digit and rounded it off. The correct figure was 0.009, as pointed out by the reviewer. There are no errors in the figures in the attached spreadsheet (CSV). Thank you for checking every detail.

Results Page 10, Table 1

precision of pVxB: 0.001 → 0.009

Comment 9:

Line 269: Please change “[…] attaching an LED […]” to “[…] attaching a LED […]”

Response: 

We thank the reviewer for this important comment. We have revised the manuscript following your comment.

Discussion Page 12, Line 270

Before: … attaching an LED …

After: … attaching a LED …

＊

The manuscript was checked by a native English speaker, and some revisions were made to the entire manuscript, such as with regard to articles ("the") and comma (", ").

Response to Reviewer #2

Comment 1:

Page 5, Line 86; check wording of this sentence. Is it true that participants performed 160 snatches at each load? The authors should reword to clarify the specific number of repetitions completed by each participant at each load.

Response:

We thank the reviewer for this pertinent suggestion. We apologize for the inadequacies of the text, as pointed out by the reviewers. We have revised the manuscript accordingly:

Materials and Methods, Page 5, Lines 87–88

Before: A total of 160 snatches were recorded 5 times for each load.

After: A total of 160 snatches were recorded, whereas every subject completed five repetitions for each load condition.

Comment 2:

Page 5, Line 93; insert ‘system’ after 3D-MC.

Response:

We thank the reviewer for this constructive comment. We have revised the manuscript following your comment.

Materials and Methods, Page 5, Line 94

Before: … 3D-MC (Vicon MX; Vicon Motion Systems, Oxford, UK).

After: … 3D-MC system (Vicon MX; Vicon Motion Systems, Oxford, UK).

Comment 3:

Page 5, Line 97; the authors state that barbell position was measured at “a higher sampling frequency”. Please clarify the exact sampling frequency to aid repeatability of the study.

Response:

We thank the reviewer for pointing this out. We have revised the sentences accordingly:

Materials and Methods, Pages 5–6, Lines 99–100

Before:

Therefore, the barbell position was measured at a higher sampling frequency to accurately measure the trajectory of the barbell including the vibration due to the collision.

After:

Therefore, the barbell position was measured at a higher sampling frequency in the present study (600 Hz) than those in the previous studies (200–250 Hz) to accurately measure the trajectory of the barbell, including the vibration due to the collision.

Comment 4:

Page 5, Line 113; insert ‘at a’ before ‘sampling’. Insert ‘using a’ before ‘shutter’. Reword ‘and white balance’ with ‘using a white balance setting’.

Response:

We thank the reviewer for this pertinent comment. We have revised the manuscript according to your comment.

Materials and Methods, Page 6, Lines 115–116

Before: … sampling rate of 100 Hz, shutter speed of 1/500 s, and white balance of 5500 K.

After: … at a sampling rate of 100 Hz, using a shutter speed of 1/500 s, and using a white balance setting of 5500 K.

Comment 5:

Page 9, Line 203; The authors state that “pVxF showed a relatively larger value than pVxB and pVy”. Please simplify this statement to enhance clarity for the reader. Is it meant that the 2D-AT method exhibited greater error for velocity measures relative to the 3D-MC method? This has not been addressed since the original review.

Response:

We thank the reviewer for this very important suggestion. We have revised the manuscript following your suggestion.

Materials and Methods, Page 9, Lines 205–207

Before:

… however, among the velocity-related variables, pVxF showed a relatively larger value than pVxB and pVy. This indicates that the precision of pVxF is lower than that of pVxB and pVy within the 2D-AT method.

After:

… however, among the velocity-related variables, pVxF showed a relatively larger value than pVxB and pVy. This indicates that pVxF has a relatively larger statistical variability in the measurement results among the variables of velocity in the 2D-AT method.

Comment 6:

Page 11, Line 242; the authors should be more concise. Consider rewording to something like ‘Therefore, it is considered that 2D-AT can obtain the barbell position coordinates with sufficient accuracy to quantify differences in the barbell displacement between successful and unsuccessful lifts as well as between different levels of athlete’.

Response:

We thank the reviewer for this valuable comment. We have revised the manuscript following your comment.

Discussion, Page 11, Lines 246–247

Before:

Therefore, it is considered that 2D-AT can obtain the barbell position coordinates with sufficient accuracy to distinguish the characteristic of the difference in the barbell displacement amount between liftings with differences between the performance level of lifting and success or failure.

After:

Therefore, it is considered that 2D-AT can obtain the barbell position coordinates with sufficient accuracy to quantify differences in the barbell displacement between successful and unsuccessful lifts as well as between different levels of lifters.

＊

The manuscript was checked by a native English speaker, and some revisions were made to the entire manuscript, such as with regard to articles ("the") and comma (", ").

---

## [Decision Letter · Decision Letter 2]

17 Jan 2022

Validation of Video Analysis of Marker-Less Barbell Auto-Tracking in Weightlifting

PONE-D-20-38952R2

Dear Dr. Nagao,

We’re pleased to inform you that your manuscript has been judged scientifically suitable for publication and will be formally accepted for publication once it meets all outstanding technical requirements.

Kind regards,

Yumeng Li

Academic Editor

PLOS ONE

Additional Editor Comments (optional):

The authors have successfully addressed reviewers' comments. The paper is accepted in its current form.

Reviewers' comments:

Reviewer's Responses to Questions

**Comments to the Author**

1. If the authors have adequately addressed your comments raised in a previous round of review and you feel that this manuscript is now acceptable for publication, you may indicate that here to bypass the “Comments to the Author” section, enter your conflict of interest statement in the “Confidential to Editor” section, and submit your "Accept" recommendation.

Reviewer #1: All comments have been addressed

2. Is the manuscript technically sound, and do the data support the conclusions?

Reviewer #1: Yes

3. Has the statistical analysis been performed appropriately and rigorously? 

Reviewer #1: Yes

4. Have the authors made all data underlying the findings in their manuscript fully available?

Reviewer #1: Yes

5. Is the manuscript presented in an intelligible fashion and written in standard English?

Reviewer #1: Yes

6. Review Comments to the Author

Reviewer #1: I would like to congratulate the authors on their article. They adressed my suggestions thoroughly and presented a final draft of the manuscript that does not raise any further questions from my end. I believe the article will be well received and complement the existing literature on automated barbell tracking.

7. PLOS authors have the option to publish the peer review history of their article (what does this mean?). If published, this will include your full peer review and any attached files.

Reviewer #1: **Yes: **Benedikt Mitter

---

## [Editor Report · Acceptance letter]

20 Jan 2022

PONE-D-20-38952R2 

Validation of Video Analysis of Marker-Less Barbell Auto-Tracking in Weightlifting 

Dear Dr. Nagao:

I'm pleased to inform you that your manuscript has been deemed suitable for publication in PLOS ONE. Congratulations! Your manuscript is now with our production department. 

Kind regards, 

on behalf of

Dr. Yumeng Li 

Academic Editor

PLOS ONE